# *Adolescer* in Time of COVID-19′s Pandemic: Rationale and Construction Process of a Digital Intervention to Promote Adolescents’ Positive Development

**DOI:** 10.3390/ijerph19052536

**Published:** 2022-02-22

**Authors:** Teresa Freire, Gabriela Santana, Alexandra Vieira, Bruna Barbosa

**Affiliations:** School of Psychology, University of Minho, 4710-057 Braga, Portugal; gabrielasantana.psi@gmail.com (G.S.); alexandra.magalhaesvieira@gmail.com (A.V.); a78495@alunos.uminho.pt (B.B.)

**Keywords:** adolescents, children, mental health, pandemic, positive development, positive psychological interventions, well-being

## Abstract

The coronavirus pandemic has severely impacted children’s and adolescents’ lives due to policies and regulations implemented to slow the virus from spreading, which led to a loss of routine, structure, academic support, and social contacts. Literature also reports a lack of outdoor activity, inappropriate diet, and disruption of sleeping habits as affecting children’s and adolescents’ lifestyles and well-being. Remarkably, these consequent psychological, behavioral, and emotional changes can compromise their self-esteem, sense of self-efficacy, and self-concept, affecting their immune systems. These maladaptive coping strategies and associated effects may emerge as a failure to access some of the sources of support that might help them cope. Facing this crisis, we aimed at promoting well-being, growth, and the positive development of Portuguese adolescents through an intervention focused on positive coping strategies. We developed “*Adolescer* in time of COVID-19—A good practices Guide for adolescents in social distancing” as a digital document to be quickly disseminated online, answering the emergent needs of Portuguese youth between 13 and 18 years old during the COVID-19 pandemic. In this article, we present the rationale and process of construction of this intervention while living within a quarantine period, considering the restrictive measures adopted at the time.

## 1. Introduction

The coronavirus pandemic severely impacts children’s and adolescents’ lives by causing high psychological pressure, with severe consequences for young people’s mental health and well-being [1,2,3]. The severe effects have been affecting everyone, either directly, through the infection of individuals or their relatives, or indirectly, through the many policies and regulations implemented to slow the virus spreading [2,3]. In the first months of 2020, during the first wave of the COVID-19 pandemic, many countries had declared unprecedented lockdowns and states of emergency, such as Portugal, whose first state of emergency was declared at the beginning of March. Schools and universities, markets, shopping centers, and other public spaces were restricted. Moreover, many events had been canceled worldwide, and families had been quarantined [4,5].

Since the declaration of the pandemic state and subsequent quarantine periods, many changes have occurred in youth’s daily life, which have caused a set of disruptions in routines, forcing them to reorganize and adapt to abrupt new life situations [5]. Being quarantined in homes have led to a loss of routine, structure, academic support, and social contacts. Other issues such as lack of outdoor activity, inappropriate diet, and sleeping habits may have happened. Such changes probably disrupted children and adolescents’ usual lifestyle, promoting monotony, distress, impatience, annoyance, and may have even provoked psychological burden [2,4,6]. In the face of isolation and lockdown measures, everyone was forced to change in innumerable ways in response to COVID-19, which created an environment of fear and anxiety [3,4,7]. This led to serious implications for children’s and adolescents’ psychological, behavioral, and emotional well-being [8,9,10], and the most adverse effects can be broad, substantial, and detected later [11]. Additionally, we suppose that such effects might be boosted by the failure to access some of the sources of support that might help them to cope. In particular, schools were closed, peer relationships were on hold or available only via screen time, and families were isolated from the community [4,6,7,8].

Recent studies revealed that the pandemic caused emotional changes in children and adolescents, causing them to feel sadness, anger, frustration, fear, or despair. Some youths also displayed disruptive behaviors, sleep disturbances, a sudden or gradual change in their eating patterns, and difficulties in talking and maintaining contact with others [4,6,8,9,10]. Socioemotional well-being is crucial for children and adolescents to develop and can have mid to long-term positive effects across the lifespan. In this particular situation, promoting well-being can help to mitigate the possible negative consequences of social isolation [10]. One of the tasks of this developmental stage is the acquisition of autonomy, which is essential for adolescents to be able to face new situations and/or tasks they might face [12]. Due to the temporary loss of autonomy and freedoms, adolescents may have experienced a lower sense of control. Individuals’ sense of control plays a key role in their immediate adjustments to the pandemic situation and their ability to shape their developmental trajectory by mediating the relationship between uncertain threats and behavioral adjustments [13].

Parents have an essential role during this situation and may help adolescents respond to the uncertainty by developing some routines and structure day by day. Additionally, parents might help by (i) providing information or clarifying youth’s doubts, which helps dealing with the situation; (ii) emphasizing that they should make the collective effort to stay safe, making sure they wash their hands and spend less time with outside family and friends; (iii) validating their feelings, and (iv) looking for ways that they can engage in helpful activities [4]. Some authors have indicated that setting times for a few regular activities each day, such as reading, playing games, home tutoring, and cooking together helps to structure a new routine and facilitate youth’s adaptation to the context of the pandemic. Moreover, activities such as videoconferencing, telephone, or real-time text messaging with friends should be considered as social interactions that are particularly important for youths. Maintaining healthy habits during this time is also crucial, so it is recommended that adolescents do some daily exercise, get fresh air while maintaining social distancing, and keep consistent sleep patterns and wake times that fit their natural rhythms [4,7]. Youth live through a crisis and experience changes in their lives, but each will experience the pandemic differently, impacting their lives in different ways [2].

This article aims to present the rationale and process of construction of a digital intervention for adolescents living in lockdown under the pandemic restrictions: The* Adolescer in time of COVID-19—A Guide for good practices for adolescents in social distancing*. Through the adverse conditions they were living in, we aimed to offer them an opportunity, through this intervention, to be involved in a series of activities focused on their personal and social development, making them capable of dealing with difficulties without compromising their health and developmental trajectories. The use of a digital intervention could match the absence of in-person interactions and relations and, thus, connect adolescents with the external world and, simultaneously, with themselves in a self-regulatory way.

## 2. Elaborating the Intervention Guide: Methods and Materials

### 2.1. Literature Support

Adolescence is a sensitive period for social development, where peer interactions become increasingly important. This peer reorientation, parallel with the development of the social brain, enable adolescents to develop into independent adults with a complete sense of social self-identity in parallel with the improvement of cognitive abilities allowing them “to better understand other people’s minds and take other’s perspectives” [14] (p. 635). Changes in social environment, as occurred within pandemic restrictions (enforced physical distancing and reduced face-to-face social contact with peers), can have detrimental effects on both brain and behavioral development. Of note is that families can play a differential role in buffering these effects. Adolescents that live with high functioning families and have positive relationships with parents/caregivers and siblings are less affected than adolescents who do not have positive family relationships or live alone [14].

These aspects integrate the Positive Youth Development (PYD) perspective that considers adolescents as agents of their own growth and development, having a natural and normative potential for healthy and prosperous development. This perspective recognizes that adolescents have strengths and skills that can be developed positively through the relational dynamics with their contexts, relationships, and interactions that constitute daily life, including the community, school, family, and peers [15,16,17]. These relational dynamics help them develop key competencies, skills, values, and self-perceptions that adaptively self-regulate individual needs to shape and navigate life over time successfully [18]. Accordingly, and looking to the pandemic period, it seemed urgent to develop strategies that could promote the positive development of adolescents through a scenario of social adversity.

Some institutions like the World Health Organization (WHO) and the Portuguese Psychologists Association (OPP) suggested, at the beginning of the pandemic, some strategies to help people, including youths, to overcome this pandemic situation as healthily as possible. In particular, OPP recommended a good practice guide for people/adults who had to be in quarantine. This Guide suggested some tips to help people during this pandemic, as (i) keeping up-to-date about the coronavirus disease; (ii) keeping in touch with family and friends; (iii) relaxing; (iv) maintaining a routine; (v) exercising; (vi) eating a balanced and healthy diet; (vii) maintaining a positive attitude, and (viii) asking for help if they feel in need [19]. As these suggestions and information came up, it was possible to verify a lack of help strategies destined explicitly for the adolescent population. With this in mind, we set out to address this gap in recommendations, developing a possible solution in terms of providing coping strategies and helping the adolescents forced into quarantine.

### 2.2. Constitution of Expert Committee

A group of nine clinical psychologists were in charge to work on the elaboration of the *Adolescer* Guide as a digital intervention. All the committee members worked with adolescents in a Psychological Association of the University (APsi–UMinho), offering services to the community. This clinical group constituted the expert committee. The coordinator of the committee (the first author of this article) conducted all the process, defining tasks, timelines and meeting schedules. The members of the committee worked on different tasks related to the following topics: (i) literature review (on adolescence/youth; Pandemic COVID-19; interventions); (ii) design and manual (e.g., figures and contents); (iii) dissemination strategies.

The committee members were ethically involved in elaborating an evidence-based intervention, contributing to the young people’s mental health in an adverse life period.

### 2.3. Expert Meetings

The expert committee members met regularly, every week, during an intensive period of one month (from 9 March to 9 April 2020, a period included in the first national lockdown). Each meeting consisted in defining the tasks for the next meeting and analyzing information and materials produced in between meetings. During the meeting sessions, all members presented the information collected or the materials produced for each session.

The meetings followed a sequence aimed at achieving the final version of the Guide of this digital intervention, which was:Literature review to validate the pertinence and relevance of the Guide for adolescents during the pandemic.Kind of contents (information, activities) to include in the Guide.Design, figures, colors, and layout of the Guide.“Thinking aloud” strategy with adolescents. Along the process of the Guide’s construction (format/layout and contents), some students/adolescents were recruited within schools to participate in this task (voluntary basis).

An agenda was systematically defined for each meeting, making possible continuous feedback on the Guide’s progress, allowing the expert group to act according to the aims of the intervention and the specified timeline. In each session, the group analyzed retrospective and prospective plans about the Guide.

### 2.4. Approval of Adolescer

The committee members approved the final version of the *Adolescer* Guide. Through group discussion and literature-based argumentation, the expert members could validate, step-by-step, all the decisions concerning format and contents of the Guide. This final version was submitted to the approval of the director of the Psychological Association of the University (APsi–UMinho).

The final *Adolescer* Guide aimed to accommodate and give answers to the emergent needs of Portuguese youth aged between 13 and 18 years old during the COVID-19 pandemic’s first lockdown. The approval was for a scientific-based intervention, supported on specific conceptual frameworks, primarily addressed for adolescents concerning their developmental characteristics in a time of public health and social adversity. The committee also approved the “hands-on” style of the *Adolescer* Guide, underlying the importance of agency in adolescence, framed by a conceptualization of development as action in context [20].

## 3. Results on the Rationale and the Construction Process

### 3.1. The “Adolescer” Conceptual Framework and Aims

The pandemic is a novel and historical event that has had an immediate impact and enduring effects on the ecological systems of everyone experiencing it [2], influencing youths’ developmental normative path. Considering Developmental Psychology, in general, and particularly the Ecological Systems Theory [19], development depends not only on the individual’s biopsychological characteristics but also on the characteristics of the surrounding contexts. Development occurs through an interrelated system of individuals and multiple settings, such as school, family, leisure, and peers. As youths interact with their immediate settings, changes that may occur within and across immediate ones will impact their development [2,21]. During the pandemic, youths’ ecological systems are in turmoil since many changes in their routines and daily habits tend to occur [2]. For example, due to permanent physical distance, adolescents were forced to readapt their interactions with friends, families, teachers, and other significant people, seeking alternative ways to communicate. Intense and abrupt changes in rules and regulations have been implemented, especially at the level of the communication patterns between people and across settings [2], affecting the ecological system of individuals, which in turn can have repercussions on their normative development.

Besides a developmental perspective, the literature review made emergent the conceptual lenses of Social Psychology (e.g., social contexts and peer social interactions), and Positive Psychology (e.g., adolescents’ strengths and assets), as well as the previously mentioned recommendations from OPP, which supported the elaboration of the intervention Guide titled *Adolescer in time of COVID-19—A good practice Guide for adolescents in social distancing*. The word *Adolescer* intends to emphasize that adolescents can continue their development healthily, despite the COVID-19 pandemic’s adverse conditions. This intervention allows for a new interpretation of recent data and findings related to this new research area of COVID-19 and its impact on adolescence, being focused on the promotion of Psychological Well-Being (PWB), growth processes, and the positive development of Portuguese adolescents. To cope with the unexpected changes, some maladaptive coping strategies may emerge showing that these are typically associated with greater levels of distress and lower PWB, while positive coping strategies are associated with greater PWB and less distress [22].

To respond to the needs and difficulties of adolescents during the pandemic (cf. previous sections of this article), we had considered the societal restrictive measures adopted at the time of the first pandemic lockdown. Literature reports that it is possible to use technology to promote adaptive behaviors and positive functioning [23]. The information and communication technologies (ICT), especially the Internet, could help to prevent and to promote positive mental health in adolescents, especially during a time when in-person interactions were suspended. The association between ICT and Positive Psychology Interventions might be an opportunity to promote youth mental health, which demonstrates important benefits in terms of sustainability and accessibility [24]. To stimulate and encourage adolescents’ adaptive coping strategies, *Adolescer* was created as a digital document that could quickly be disseminated online, reaching schools and other public and community settings in a period of severe lockdowns.

### 3.2. The Intervention Guide for Adolescents in Quarantine: The “Adolescer in Time of COVID-19”

Focusing on the Portuguese context, *Adolescer* adapted OPP’s eight suggestions to deal with social distance within pandemic restrictions, and considered adolescents’ characteristics and the developmental period they were going through. These suggestions were: (i) “keep yourself updated”; (ii) “keep in touch”; (iii) “relax”; (iv) “keep a daily routine”; (v) “practice physical exercise”; (vi) “maintain a healthy diet”; (vii) “stay positive”, and (viii) “ask for help when needed”.

#### 3.2.1. Literature Review on Main Intervention Strategies

We based the adaption of the eight suggestions to the adolescent population on specific literature that could support eight main intervention strategies to apply to adolescents. The literature review supporting each one of the strategies included in the Guide is presented next.

##### “Keep Yourself Updated”

In unknown situations that provoke high stress, such as the pandemic, it is important to regulate the information sources as well as the frequency in which adolescents listen to that information. For adolescents, this is highly prevalent as they like searching information on the internet, or sharing it between themselves (peers), most of the time without any serious filter on credible information sources. Media can create the idea that there is a greater and permanent danger and risk than what exists [25]. Watching, reading, or listening to the news several times a day may cause greater anxiety, distress, or worries [26]. In order to prevent this, some health organizations recommended trying to keep up to date with what was going on, while limiting the exposure to the news [25].

Following this information, we also advise adolescents to seek information updates on a fixed timetable, a maximum of once or twice a day, using websites from official institutions, such as the Portuguese health committee (Direção Geral de Saúde—DGS) and WHO [25,26]. We believe it is important that adolescents keep themselves up to date, while being able to distinguish facts from rumors or fake news [26].

##### “Keep in Touch”

As literature demonstrates, adolescence is denoted by profound physical and psychosocial changes that assume extreme relevance in the sphere of interpersonal relationships [27]. This stage is characterized by a progressive transition, from the amount of time spent with their parents to the development of more complex relationships with peers, as a consequence of more time spent with them [28]. Peer relationship quality, therefore, impacts psychological well-being, the development of behavioral and emotional competencies, as well as adolescents’ subsequent adjustment to adulthood’s social interactions [27,29], protecting them from mental health problems and strengthening their resilience [30].

Although the quality of peer relationships assumes extreme importance in adolescence, so do family relationships. Adolescents that maintain better relationships with parents and siblings might suffer fewer consequences derived from social distancing [14]. Bearing this in mind, as well as the fact that maintaining social contact through digital media is believed to ease the potential consequences of social distancing in youths [14], we thought of some activities which they could develop.

Thus, our suggestions focused on challenging adolescents to interact with their families through board games or the creation of novel activities to develop at home, as well as suggesting interactions with their peers through video chats, online study groups, and leisure activities that took advantage of various digital apps available for movies, exercise or online games (e.g., “Videochat with your family, friends/colleagues, and teachers through the various available apps.”).

##### “Relax”

During their normative transition into adulthood, adolescents’ may demonstrate feelings of stress related to school, relationships, and family, which can take the form of maladaptive coping strategies [31], or even somatic symptoms (e.g., headaches or low energy [32]). Besides these stressors normally associated with the adolescence developmental phase, lockdown conditions derived from the pandemic should be considered as a possible stressor for this age group. These may produce continuous stress responses limited in time (i.e., tolerable stress [33]), which could cause a bigger impact on adolescents’ development and their social, emotional, and behavioral functioning [31]. Stress responses can be reduced, and the previously mentioned effects reversed, by the existence of responsive and stable relationships [33], as well as by the implementation of relaxation and stress management strategies that show a positive impact on psychological well-being [22,31].

Based on this information, our suggestions for this topic focused on teaching adolescents on how to engage in diaphragmatic breathing and progressive muscle relaxation, encouraging them to practice, and consequently learning how to deal with stressors, and relax (e.g., “Breathing and relaxation techniques are great tools to help decrease anxiety and stress levels. Follow our instructions and relax…”).

##### “Keep a Daily Routine”

As mentioned before, adolescence is a development phase denoted by many changes, including those in sleep patterns. These are associated with biological shifts in the circadian cycles towards a preference for later bed and wake times (i.e., evening types [34,35]) and also by environmental factors, such as less parental control and electronic media use [36,37]. With schools closing and adolescents being confined at home due to the pandemic COVID-19 restrictions, daily routines were significantly altered. Recent studies demonstrate that the regularity of daily routines during the pandemic lockdown period moderated the effects of the pandemic on changes in mental health. Particularly, family routines are described as significant factors in adolescents’ development and contribute to their well-being in high-stress situations [38,39]. Additionally, most of their activities and ways of keeping in touch with friends/colleagues, family, and teachers were only possible through electronic media. Literature shows that evening tendency and more screen time affect bedtime and sleep duration. Reduced family routines, specifically bedtime and screen time routines, during COVID-19 may negatively affect adolescents’ health behaviors [40]. This reduction impacts adolescents’ daily functioning in terms of more sleepiness, problems related to falling asleep and waking up, and more fatigue [35,36], contributing to attention problems, lower academic productivity/achievement, behavioral and emotional problems, [41] and decreases in subjective well-being [42]. Although family routines were difficult to maintain in the context of the pandemic, they were associated with better individual and family well-being during this period of acute health, economic, and social stress [40].

These changes impact adolescents’ well-being and productivity, supporting the need to encourage them to define timetables and sleeping hours and screen time activities, along with regular physical exercise and hours for their meals (as will be discussed later in this text). Following these aspects, we challenged adolescents to “Build a daily or weekly plan with clear and specific objectives that you think you can accomplish. Define hours for your sleep, meals, study, and leisure”, thus keeping them engaged and connected with daily life tasks.

##### “Practice Physical Exercise”

During pandemic restrictions, adolescents and their families were limited to the space of their homes which, in turn, has direct consequences in the decrease of physical activity (e.g., due to reduced space). This is conversely associated with an increase in sedentary behaviors, characterized by bigger amounts of time spent sitting or lying down watching screens (e.g., computer), which represents negative consequences for physical and mental health (e.g., risks for chronic diseases), sleep patterns, well-being, and quality of life [43,44]. As evidence, implementation of physical activity in the current lockdown conditions is of extreme importance, not only in terms of physical health (e.g., maintenance of muscular functions [44]) but also on a psychological level (e.g., adolescents who frequently engage in physical activity present more protective factors related to resilience [45]). Ensuring that adolescents exercise at home also contributes to avoiding the emergence of non-healthy habits, maintaining pre-existing healthy habits, and promoting new ones that guarantee better resilience when lockdown ends [44].

Following these considerations, the activity we suggested for this topic encourages adolescents to exercise by themselves, or accompanied by their family members, for a minimum of 30 min per day using digital apps, or even practicing domestic chores, ensuring that, by doing so, there would be a decrease in sedentary behaviors (e.g., “Organize your weekly timetable so that it fits at least 30 min of physical activity per day. Don’t forget that articulating physical activity with study time will positively impact your productivity!”).

##### “Maintain a Healthy Diet”

While physical activity decreases, the growth of sedentary behaviors is also associated with changes in eating behaviors and higher energy intakes [44]. Literature shows that sedentary behaviors related to the constant use of screen devices are usually associated with an increase in lower nutritional food, such as snacks, sweets, and fried food [46]. The adoption of non-nutritional/non-balanced diets could negatively impact levels of vitamin D, which could already be affected due to lockdown and lack of outdoor activities [44]. It is, then, important to promote and ensure that adolescents maintain balanced diets to prevent possible irreversible consequences derived from non-healthy eating habits associated with sedentary behaviors [44].

To promote healthy eating habits and show adolescents that food can be fun and exciting, we challenged them to cook at least one of the recipes we made available for them. The activity for this topic intended to encourage adolescents to look for food alternatives and try them, (e.g., “We challenge you to cook at least one of these recipes that prove healthy is tasty!”) as well as to manage meal times and avoid possible irregular food intake.

##### “Stay Positive”

Positive emotions work as buffers for the adverse effects of negative emotions, assuming an active role as promoters of resilience and well-being [47], namely in post-crisis situations [48]. According to Fredrickson [49], positive emotions also impact broadening people’s attention and thinking, and contribute to the acquisition of personal resources, which can be useful as future coping and survival strategies, especially important for adolescents. Due to all the changes caused by the pandemic and the necessity of home confinement, it is necessary to support adolescents in staying positive and not to focus only on the negative situations and emotions they might be experiencing.

Accordingly, our activity suggestion focuses on adolescents’ capacity to adapt and flourish through adversity by staying positive and identifying which personal resources and competencies they possess, and which ones can be used to improve their days, and those of the ones around them (e.g., “Have you thought how your skills can contribute to making your days and those of the ones around you better while dealing with social distancing?”).

##### “Ask for Help When Needed”

As important as it is to recognize personal competencies and apply them to cope with difficult situations, the absence of coping mechanisms must also be recognized.

Asking for help with trusting professionals in situations where there are no available personal resources is also a relevant coping strategy, especially in adolescence, since this is a stage of great change and self-knowledge. Adolescents facing new and unexpected situations, such as the COVID-19 pandemic, might not know how to act and deal with their feelings. Encouraging them to ask for help is the main strategy to promote this valued strategy. To be able to ask for help, adolescents need to know which external resources are available to them. This way, the activity suggested for this topic focused on showing adolescents which entities they could get in contact with, and also encourages them to search for and register other means available to them when needed (e.g., It is very important that you are aware of the resources available to help you in necessary situations. Research helplines are available in your residency area and register them so you can keep the information and/or share it with your family and/or friends.”).

### 3.3. The “Thinking Aloud” Strategy with Adolescents

This strategy is widely used when it is necessary to know about thoughts, feelings, and impressions of a specific population [50] in order to elaborate or adapt any kind of materials to that population. Usually this strategy constitutes an adequate source of information to elaborate or adapt new materials (written or visual). In line with this, we included some adolescents along the process of construction of the Adolescer Guide. On a random basis, we applied the strategy to two adolescents (male and female) available to analyze the materials of the Guide. Three members of the expert committee were in charge of this task, asking adolescents to say out loud what they were thinking/feeling about the materials we presented them. They could speak about any issue related to the materials and all their verbalizations were registered. Their comments, impressions, and ideas were integrated along the process of construction. One example of questions was about the figures of individuals and the absence of characteristics like faces, hair, clothing. They were particularly attentive to this and said and agreed “it was a good idea because like this, we can all identify with these figures”.

### 3.4. The Final Structure of the “Adolescer” Guide

We defined the final structure of the *Adolescer* Guide considering two main aspects: (i) the broad conceptual frameworks and specific literature supporting the main strategies to understand adolescents’ positive development requirements while living in a health pandemic quarantine; (ii) the feedback provided by adolescents when “thinking aloud” about materials and formats of the *Adolescer* Guide.

Aiming to achieve our primary goal of promoting youth’s well-being and positive development by fostering coping strategies to deal with the COVID-19 pandemic, the *Adolescer* Guide followed an organized structure. Besides challenging adolescents to execute and implement the activities created for them, it was also important to let them know why these main strategies became relevant to cope with the pandemic healthily. Therefore, and in order to integrate all these issues, we decided to divide our Guide into three parts.

In the first part, the main eight strategies were introduced in order to let adolescents know how they could cope with the implemented lockdown measures. These were adapted to the *Adolescer* Guide by being presented in direct speech along with a representative image for each strategy. Each of these images was carefully thought about, displaying elements and activities easily recognizable on their own, accounting for adolescents’ sense of representation, as the figures chosen did not specify gender or physical characteristics—see Figure 1.

In a second part, bearing in mind the importance of transmitting to adolescents how essential it was for them to follow each suggestion, underlying the benefits associated with them, all the previously gathered data were converted into “Did you know…?” curiosities. These were accompanied by the figures chosen for each strategy, as well as the indication to implement the activity outlined for that topic—see Figure 2.

Lastly, the third part was entirely dedicated to the activities previously created for each of the eight suggestions that constitute the “*Adolescer*” Guide. Although each topic only had one activity associated with it, some activities combined more than one “challenge”, encouraging adolescents to implement them, either individually or with their families or friends, following the implemented restrictions. It is also relevant to mention that, depending on the type of activity or challenge outlined, sheets with entry fields were made available, where adolescents were able to write and register the exercises they completed—see Figure 3.

As a bonus activity that integrated all the suggestions in the *Adolescer* Guide, a “Final Challenge” was created, which adolescents could choose to use as a calendar or a board game. The former was meant to be implemented individually, allowing adolescents to accomplish thirty daily challenges over a month. The latter could be played along with family or friends, following the ground rules described in the Guide—See Figure 4.

Finally, to make sure adolescents were aware of the constant updates and new information that emerged weekly about the coronavirus disease and its worldwide evolution, a series of social media accounts were suggested in the *Adolescer* Guide. These belonged to official institutions such as “Direção Geral de Saúde” (DGS) or the WHO, which allowed youths to have access to reliable information.

### 3.5. Dissemination of the Digital Guide Adolescer

The “*Adolescer*” Guide (in the format of a digital document) was made available online through social media networks (formal and informal) and institutional sites (APsi-UMinho) to easily diffuse through the population, ultimately reaching the target age group.

In addition, we sent the Guide to a pool of national schools (directly to their directors), asking for its ethical dissemination amongst students, and to other national youth organizations, trying to reach adolescents directly or through stakeholders.

## 4. Discussion

COVID-19 has caused several changes in adolescents’ daily lives, which required them to adapt. The current pandemic reinforced the importance of interventions in crisis and the need for a quick and effective response to facilitate the adaptation of adolescents to abrupt changes in their lives and the continuous promotion of their positive development instead. Adolescence is characterized by remarkable changes that make emergent new skills and competencies that potentiate adolescents and, at the same time, can cause stressful situations because of novelty in daily life, unknown consequences, and/or the complex challenge of knowing themselves [51]. Adverse conditions caused by the coronavirus disease can disrupt adolescents’ lives, which, if not well managed, can trigger psychological disorders, compromising their mental health and optimal functioning in daily life and throughout life.

### 4.1. The Meaning of Adolescer Development

*Adolescer* is a verb that means to enter in adolescence and flourish. Attending to the pandemic situation, our first insight to start the elaboration of this intervention in the middle of the pandemic was to guarantee that adolescents could continue their flourishing process while living in a pandemic with severe restrictions to their developmental sources.

Therefore, *Adolescer in time of COVID-19—A good practice Guide for adolescents in social distancing*, was created to give healthy strategies and facilitate a proactive adaptation and reorganization of daily life in order to promote healthy development, answering the needs of adolescents going through social distancing in pandemic lockdowns. Due to its evidence-based features, and following guidelines from serious and credible organizations (e.g., OPP), this Guide (preliminary version) emerged as an adequate digital tool to be easily used by adolescents, even if in social distancing.

### 4.2. Limitations and Future Research

The construction process of this intervention was a big challenge considering the social conditions at the beginning of the COVID-19 pandemic. Even working in the middle of the pandemic, the expert committee aimed to give fast support to adolescents throughout their daily lives. The urgency of the need for this intervention was its most significant limitation, as no time existed to consider a long process to validate the intervention before its implementation. Only an empirically-based structure, supported in a strong conceptual rationale, could overcome that limitation. The severe disruption in adolescents’ daily lives was the most comprehensive evidence-based fact to keep this digital intervention’s continuity.

We intend to continue this *Adolescer* project, trying to adapt and implement it concerning the societal changes and mental health issues emerging along with the pandemic and in a post-pandemic future. New design assessment plans using quantitative (retrospective self-reports and questionnaires) and qualitative methods (case studies with adolescents that used the Guide) will be provided to evaluate its efficacy in terms of benefits, thus producing a final manual able to be used by adolescents or by education stakeholders (teachers, parents, community managers) in their relations with adolescents in daily life.

## 5. Conclusions

In a period of constant changes and successive adaptations related to a pandemic, digital intervention tools emerge as an effective and accessible way to boost adolescents when dealing with daily life adversities, as is the case of the COVID-19 pandemic. The novelty of this pandemic gives us the possibility of knowing new and different variables on human functioning following an ecological framework, which will provide profound and productive contributions to the understanding of adolescents’ flourishing and thriving.

## Figures and Tables

**Figure 1 ijerph-19-02536-f001:**
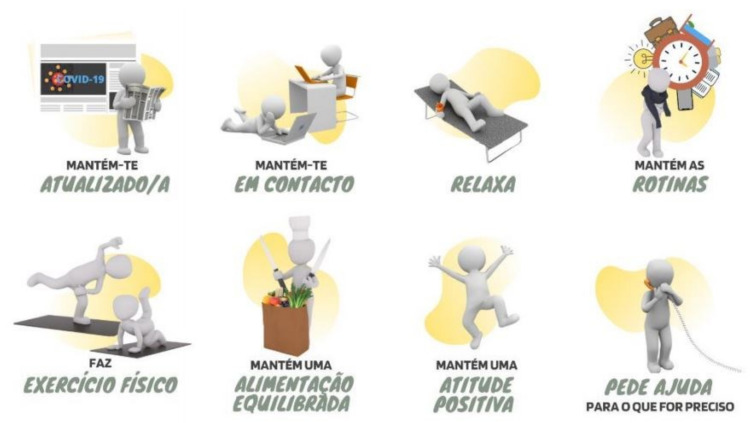
Intervention strategies of the *Adolescer* Guide. Note: The image shows the original page from “Adolescer”, representing the eight practices written in Portuguese. Following the order from left to right, in English the figures translate to: (i) “keep yourself updated”; (ii) “keep in touch”; (iii) “relax”; (iv) “keep a daily routine”; (v) “practice physical exercise”; (vi) “maintain a healthy diet”; (vii) “stay positive” and (viii) “ask for help when needed”.

**Figure 2 ijerph-19-02536-f002:**
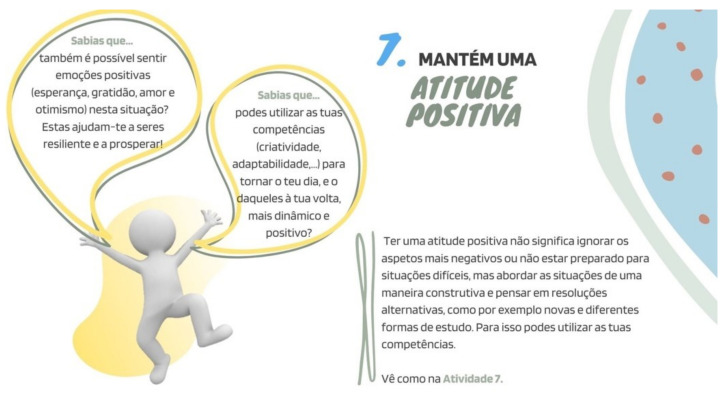
Example of informative contents for a specific strategy (activity 7: “keep a positive attitude”).

**Figure 3 ijerph-19-02536-f003:**
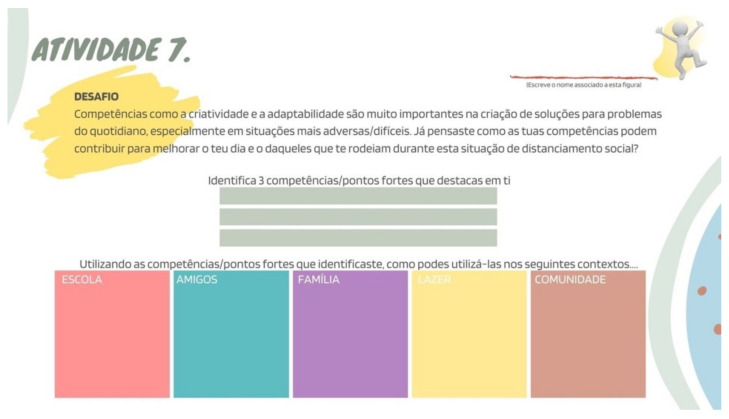
Example of suggested activity.

**Figure 4 ijerph-19-02536-f004:**
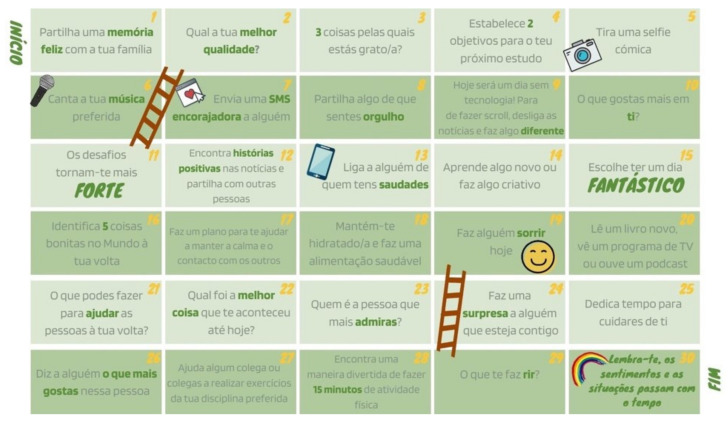
The final integrative activity.

## Data Availability

Not applicable.

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
