# Peer review of "Adolescer in Time of COVID-19′s Pandemic: Rationale and Construction Process of a Digital Intervention to Promote Adolescents’ Positive Development"

_ijerph, 2022, doi:10.3390/ijerph19052536_

Round 1
Reviewer 1 Report
I read with great interest the article entitled "Adolescer” in Time of COVID-19’s pandemic: A preliminary digital intervention to promote adolescents’ healthy and positive development" where the authors present the theoretical and methodological path they followed to develop “Adolescer” in time of COVID-19 – a good practices guide for adolescents in social distancing”, as a digital document to be quickly disseminated online, answering the emergent needs of Portuguese youth between 13 and 18 years old during the COVID-19 pandemic.
I am aware that as a result of the covid-19 pandemic, adolescents have unfortunately been almost forgotten by institutions. The negative effects not only of the pandemic, but also of the lack of interest shown towards young people, can be assessed in the near future by monitoring, for example, the possible increase in the number of accesses to mental health services, the decrease in school performance, etc...
I believe that this work may be worthy of publication because it presents the implementation of one of the few interventions aimed at adolescents in order to support them during the pandemic.
For the purposes of publication, however, I think the article could be improved if the authors were able to respond to the following comments:
1-In the text, the authors often claim that relationships with peers during adolescence are very important (e.g. page 2 they say “In adolescence, relationships with peers as-88 sume extreme importance, although family relationships are still crucial for the healthy 89 development of youth [18].” and then again on page 3and on page 3 “From a social psycholog-139 ical point of view, adolescence is characterized by a heightened need for peer interaction 140 [18], fundamental for the healthy development of adolescents [22,23,18].”), but they never go into any depth in the direction of giving an explanation for such claims.
I think the authors can refer to Sherif & Sherif's concept of "social laboratory", illustrating how during adolescence the peer group plays an important role in accompanying the adolescent in the process of autonomisation and de-individuation from parental figures.
2- On p. 6, the authors discuss the importance of maintaining a daily routine for the promotion of one's well-being, but conceptualising this in purely physiological terms (e.g. duration and quality of sleep and its impact on the productivity of daily tasks).
There is a line of research in the literature showing that daily routines can play an important role as a protective factor against psychosocial risk (e.g. see Fiese 1992, 2006; Emiliani et al. 1998, 2007; Emiliani 2009; Migliorini et al 2011; Lagomarsino et al. 2020).
I believe that in section 3.1.4 a more in-depth discussion of this issue would benefit the quality of the article.
Having settled these minor remarks, I consider the work worthy of publication, Congratulations to the authors.
References
Emiliani, F. (2009). A Realidade das Pequenas Coisas: a Psicologia do Cotidiano. Senac: São Paulo.
Emiliani, F., Melotti, G., Palareti, L. (1998). Routine e rituali della vita familiare quali indicatori di rischio psicosociale. Psicologia Clinica dello Sviluppo, 2(3), pp. 421-448
Emiliani, F., Melotti, G., Palareti, L. (2007). Social representations of everyday life and well-being in Italian adolescents | [Représentations sociales de la vie quotidienne et bien-être chez des adolescents italiens]. Revue Internationale de Psychologie Sociale, 20(2), pp. 27-55
Fiese, B.H. (1992). Dimensions of Family Rituals across two generations: Relations to adolescent identity, Family Process, 31, 151-162.
Fiese, B.H. (2006). Family routines and rituals. Yale University Press
Lagomarsino, F., Coppola, I., Parisi, R., Rania, N. (2020). Care Tasks and New Routines for Italian Families during the COVID-19 Pandemic: Perspectives from Women. Italian Sociological Review; 10(3S), 847-868,847A. DOI:10.13136/isr.v10i3S.401
Migliorini, L., Cardinali, P., Rania, N. (2011). La cotidianidad de lo familiar y las habilidades de los niños. Psicoperspectivas (Valparaiso), 10 (2), p.183-201
Author Response
Please see my answers to Reviewer 1
Reviewer - For the purposes of publication, however, I think the article could be improved if the authors were able to respond to the following comments:
1-In the text, the authors often claim that relationships with peers during adolescence are very important (e.g. page 2 they say “In adolescence, relationships with peers as-88 sume extreme importance, although family relationships are still crucial for the healthy 89 development of youth [18].” and then again on page 3and on page 3 “From a social psycholog-139 ical point of view, adolescence is characterized by a heightened need for peer interaction 140 [18], fundamental for the healthy development of adolescents [22,23,18].”), but they never go into any depth in the direction of giving an explanation for such claims.
ANSWER: Thank you for your comment. We agreed and have developed the text in order to give an "explanation for such claims" - We have changed some order in the text also trying to synthesize.
Reviewer- I think the authors can refer to Sherif & Sherif's concept of "social laboratory", illustrating how during adolescence the peer group plays an important role in accompanying the adolescent in the process of autonomisation and de-individuation from parental figures.
ANSWER: We thank you for your suggestion. Sherif & Sherif's concept is relevant for this topic, but in order to make more salient these aspects in relation to the pandemic, we decided to support our information on these articles that are focused in the actual pandemic and discuss these developmental aspects accordingly. We think there are already enough references along the text about peer group...
Reviewer - 2- On p. 6, the authors discuss the importance of maintaining a daily routine for the promotion of one's well-being, but conceptualising this in purely physiological terms (e.g. duration and quality of sleep and its impact on the productivity of daily tasks).There is a line of research in the literature showing that daily routines can play an important role as a protective factor against psychosocial risk (e.g. see Fiese 1992, 2006; Emiliani et al. 1998, 2007; Emiliani 2009; Migliorini et al 2011; Lagomarsino et al. 2020).
ANSWER - Thank you. We agreed with your suggestions and we changed accordingly. We use some of your suggestions for references, and added in the text more information from these articles.
Reviewer - I believe that in section 3.1.4 a more in-depth discussion of this issue would benefit the quality of the article.
ANSWER - Done.
Having settled these minor remarks, I consider the work worthy of publication, Congratulations to the authors.
Reviewer 2 Report
The COVID-19 pandemic introduced a lot of burden for all members of the societies, including children and adolescents. Effective interventions are important for protecting persons from adverse effects of pandemic related restrictions. Therefore the manuscript warrants attention. Nevertheless in its present form the manuscript offers nothing more than the description of the intervention package and its rationale, the general conclusions of its effectiveness (lines 418-419) and, some general information that the intervention is being tested (lines 420-424). Although the "Adolescer" decribed in the manuscript is interesting and based on sound assumptions, but without any information concerning the procedure of its evaluation it is difficult to agree with the Authors who claim its effectiveness (line 149). Therefore it is important to elaborate on the mansucript and include some details related to evaluation of the procedure and the reception of "Adolescer" among stakeholders, for example: how many schools/individuals were using it, what criteria of effectiveness were applied, what are the preliminary results of such evaluation. I would also suggest the Authors to revise the text with the purpose to avoid repetition of information concerning the adolescence period (such information is given both in the 1 and 2 section of the text).
Author Response
Reviewer - The COVID-19 pandemic introduced a lot of burden for all members of the societies, including children and adolescents. Effective interventions are important for protecting persons from adverse effects of pandemic related restrictions. Therefore the manuscript warrants attention. Nevertheless in its present form the manuscript offers nothing more than the description of the intervention package and its rationale, the general conclusions of its effectiveness (lines 418-419) and, some general information that the intervention is being tested (lines 420-424). Although the "Adolescer" decribed in the manuscript is interesting and based on sound assumptions, but without any information concerning the procedure of its evaluation it is difficult to agree with the Authors who claim its effectiveness (line 149).
ANSWER - Thank you for your comment. The main aim of the article is to present the elaboration process of the intervention "Adolescer". Therefore we do not present the design of evaluation. But in order to make this clearer in the text, we change some parts along the text and add some more information on the elaboration process, also referring this aspect in the discussion.
Reviewer - Therefore it is important to elaborate on the mansucript and include some details related to evaluation of the procedure and the reception of "Adolescer" among stakeholders, for example: how many schools/individuals were using it, what criteria of effectiveness were applied, what are the preliminary results of such evaluation.
ANSWER: Although we do not have all these data, we add more information on these aspects in new parts of the manuscript - see points 2. (and inside this, the subsections), 3. and 4. in order to clarify these aspects about evaluation and elaboration of the interventions under specific conditions.
Reviewer - I would also suggest the Authors to revise the text with the purpose to avoid repetition of information concerning the adolescence period (such information is given both in the 1 and 2 section of the text).
ANSWER: Thank you for the suggestions. We totally agreed and change the text accordingly. We revised all the text and deleted repetitions and or change some parts of the text into other parts. We think now the text is more balanced in terms of extension of the text by the different parts, which has solved the majority of the repetitions.
Reviewer 3 Report
The authors developed "Adolescer-A good practices guide for adolescents in social distancing". However, the manuscript must be revised or supplemented for clarity.
(Comment 1) The authors developed "Adolescer". But, development method (e.g. literature review, consist of expert committee, expert meeting, and approval or consensus of Adolescer) was not described in the paper. I recommend authors to supplement "Materials and Methods Section".
e.g.
- Introduction
- Materials and Methods
2.1. Literature review
2.2. Consist of expert committee
2.3. Expert meeting
2.4. Approval of Adolescer
- Results
3.1. The “Adolescer” Conceptual Framework and Aims
3.2. The Intervention Guide for Adolescents in Quarantine: The “Adolescer in Time of 185
COVID-19”
- Dicussion
- Conclusion
Discussion Section
(Comment 2) In the "Discussion Section", the following should be discussed. I recommend authors to re-write the "Discussion Section"
(1) The meaning of "Adolescer" development
(2) Limitation of this research
(3) Application strategy of "Adolescer"
(4) Suggestion for future research
(Comment 3) I recommend authors to revise the following points.
(line 46) [2,6,4] -> [2,4,6]
(line 48) [7,4,3] -> [3,4,7]
(line 54) [8,6,7,4] -> [4,6-8]
(line 59) [8,6,4,9,10] -> [4,6,8-10]
(line 106) [7,4] -> [4,7]
(line 141) [22,23,18] -> [18,22,23]
(line 147) [24,16] -> [16,24]
(line 245) [32,25] -> [25,32]
(line 296) 3.1.5.“. Practice Physical Exercise” -> 3.1.5.“Practice Physical Exercise”
(line 275) 3.1.6.“. Maintain a Healthy Diet” -> 3.1.6.“Maintain a Healthy Diet”
Author Response
The authors developed "Adolescer-A good practices guide for adolescents in social distancing". However, the manuscript must be revised or supplemented for clarity.
(Comment 1) The authors developed "Adolescer". But, development method (e.g. literature review, consist of expert committee, expert meeting, and approval or consensus of Adolescer) was not described in the paper.
REVIEWER - I recommend authors to supplement "Materials and Methods Section".
ANSWER - Done. We follow all your suggestions and did all the changes suggested:
e.g.
- Introduction
- Materials and Methods - Done
2.1. Literature review - Done
2.2. Consist of expert committee - Done
2.3. Expert meeting - Done
2.4. Approval of Adolescer - Done
3. Results - Done
3.1. The “Adolescer” Conceptual Framework and Aims - Done
3.2. The Intervention Guide for Adolescents in Quarantine: The “Adolescer in Time of COVID-19” - Done
- Dicussion - Done
- Conclusion - Done
Discussion Section - Done
(Comment 2) In the "Discussion Section", the following should be discussed. I recommend authors to re-write the "Discussion Section"
(1) The meaning of "Adolescer" development - Done
(2) Limitation of this research - Done, but we join this point 2 with point 4 - with the title of Limitations and future research
(3) Application strategy of "Adolescer" - we didn't use this point because we think it is already explained in the text
(4) Suggestion for future research - Done (with point 2.)
(Comment 3) I recommend authors to revise the following points. - all done
(line 46) [2,6,4] -> [2,4,6]
(line 48) [7,4,3] -> [3,4,7]
(line 54) [8,6,7,4] -> [4,6-8]
(line 59) [8,6,4,9,10] -> [4,6,8-10]
(line 106) [7,4] -> [4,7]
(line 141) [22,23,18] -> [18,22,23]
(line 147) [24,16] -> [16,24]
(line 245) [32,25] -> [25,32]
(line 296) 3.1.5.“. Practice Physical Exercise” -> 3.1.5.“Practice Physical Exercise”
(line 275) 3.1.6.“. Maintain a Healthy Diet” -> 3.1.6.“Maintain a Healthy Diet”
Round 2
Reviewer 2 Report
The Authors have considered my comments to the previous version of the manuscript. Thank you!
Implemented changes clarify the process of "Adolescer" construction and provide additional important information. Although it may be difficult to accept additional comments related to the ammended manuscript, but I would like the Authors to consider the following:
1) In Section 2.4 only the approval from the committe memembers is mentioned. Where there any external bodies involved? What was the opinion of adolescent experts who took part in "thinking aloud" strategy (section 2.3)? In my opinion adolescent experts views should be clearly presented, as they represent the target population for "Adolescer".
2) In lines 237-238 - please clarify which part of the text are referred to when sections are mentioned.
Although the Authors did not provided any information concerning the evaluation of their intervention programme but included comments with the rationale why such evaluation was not possible. It is not the best but at least acceptable solution. To reflect that I would also suggest the small change in the title:
3) adding the phrase "the rationale and the process of construction" at the end of the title.
Author Response
Thank you for your comments and suggestions.
Please see our answers to your comments, point by point:
Reviewer:
1) In Section 2.4 only the approval from the committe memembers is mentioned. Where there any external bodies involved?
Author: Yes we added this information, clarifying the approval process – see section 2.4
Reviewer: What was the opinion of adolescent experts who took part in "thinking aloud" strategy (section 2.3)?
Author: The strategy of "thinking aloud" was used with adolescents. It is a strategy of asking people from the population with whom we want to implement the intervention. Two adolescents were recruited to analyze materials and talk about what they think. This was also added as a new section inside the section of results, clarifying this process – please see section 3.3 (besides the information on section 2.3).
Reviewer: In my opinion adolescent experts views should be clearly presented, as they represent the target population for "Adolescer".
Author: These “adolescents experts” as you are considering do not exist in our intervention process. Please see the answer to previous comment about the thinking aloud strategy.
Reviewer: 2) In lines 237-238 - please clarify which part of the text are referred to when sections are mentioned.
Author: I think this is done.
Reviewer: Although the Authors did not provide any information concerning the evaluation of their intervention programme but included comments with the rationale why such evaluation was not possible. It is not the best but at least acceptable solution. To reflect that I would also suggest the small change in the title, 3) adding the phrase "the rationale and the process of construction" at the end of the title. :
Author: We totally agreed about changing the title. In line with your suggestion we changed it, but in a different way because of the extension of the title. So for the final title, we deleted the word "healthy" because it is included in the concept of positive development, and once the title has two parts, we put the new part of the title at the beginning of the second part. we hope you agree. The final title is then: Adolescer in Time of COVID-19’s Pandemic: Rationale and Construction Process of a Digital Intervention to Promote Adolescents’ Positive Development
Reviewer 3 Report
After reviewing the author responses, the author successfully addressed most of the comments and suggestions.
Author Response
Following the notification of this reviewer about English:
(x) Moderate English changes required
We reviewed the text, changing some words or sentences along the text in English.
Thank you.